# Meta-Analysis of COVID-19 Metabolomics Identifies Variations in Robustness of Biomarkers

**DOI:** 10.3390/ijms241814371

**Published:** 2023-09-21

**Authors:** Anthony Onoja, Johanna von Gerichten, Holly-May Lewis, Melanie J. Bailey, Debra J. Skene, Nophar Geifman, Matt Spick

**Affiliations:** 1School of Health Sciences, Faculty of Health and Medical Sciences, University of Surrey, Guildford GU2 7XH, UK; a.onoja@surrey.ac.uk (A.O.); n.geifman@surrey.ac.uk (N.G.); 2School of Chemistry and Chemical Engineering, Faculty of Engineering and Physical Sciences, University of Surrey, Guildford GU2 7XH, UK; j.vongerichten@surrey.ac.uk (J.v.G.); m.bailey@surrey.ac.uk (M.J.B.); 3School of Biosciences, Faculty of Health and Medical Sciences, University of Surrey, Guildford GU2 7XH, UK; holly-may.lewis@surrey.ac.uk (H.-M.L.); d.skene@surrey.ac.uk (D.J.S.)

**Keywords:** COVID-19, diagnostics, machine learning, mass spectrometry, metabolomics, validation, biomarkers, future pandemics

## Abstract

The global COVID-19 pandemic resulted in widespread harms but also rapid advances in vaccine development, diagnostic testing, and treatment. As the disease moves to endemic status, the need to identify characteristic biomarkers of the disease for diagnostics or therapeutics has lessened, but lessons can still be learned to inform biomarker research in dealing with future pathogens. In this work, we test five sets of research-derived biomarkers against an independent targeted and quantitative Liquid Chromatography–Mass Spectrometry metabolomics dataset to evaluate how robustly these proposed panels would distinguish between COVID-19-positive and negative patients in a hospital setting. We further evaluate a crowdsourced panel comprising the COVID-19 metabolomics biomarkers most commonly mentioned in the literature between 2020 and 2023. The best-performing panel in the independent dataset—measured by F1 score (0.76) and AUROC (0.77)—included nine biomarkers: lactic acid, glutamate, aspartate, phenylalanine, β-alanine, ornithine, arachidonic acid, choline, and hypoxanthine. Panels comprising fewer metabolites performed less well, showing weaker statistical significance in the independent cohort than originally reported in their respective discovery studies. Whilst the studies reviewed here were small and may be subject to confounders, it is desirable that biomarker panels be resilient across cohorts if they are to find use in the clinic, highlighting the importance of assessing the robustness and reproducibility of metabolomics analyses in independent populations.

## 1. Introduction

COVID-19, first reported in 2019, rapidly spread globally and led to the declaration of a pandemic by the World Health Organization (WHO) on 11 March 2020 [1]. The disease has since affected millions of people worldwide with a significant impact on public health and welfare [2]. In the initial stages of the pandemic, there was an insufficiency of testing, which led to research into the possibility of mass spectrometry-based testing as an alternative means of detection [3,4]. Later, as reverse transcription polymerase chain reaction (RT-PCR) and rapid antigen testing (RAT) capacity increased dramatically, the focus of metabolomics work shifted to identifying dysregulated pathways and markers for patient vulnerabilities and potential treatment candidates. Whilst many metabolomics studies have been conducted in a variety of matrices including breath [5], sebum [6], urine [7], and saliva [8], the majority of tests have used peripheral blood as a sampling matrix [9,10]. Whilst more invasive than breath or sebum, for example, peripheral blood has the advantage of being homeostatically regulated and less subject to contamination.

To date, metabolomic biomarkers for COVID-19 in both sera and plasma have mainly fallen into two categories. A number of studies have focused on the dysregulation of amino acids [11,12]. In part, this dysregulation may also be derived from inflammation, given that elevated levels of pro-inflammatory cytokines, such as interleukin-6 (IL-6), have been associated with alterations in amino acid metabolism [13]. Other studies have identified changes in lipid expression in COVID-19-positive patients [14,15,16]. COVID-19 triggers an immune response, leading to an inflammatory reaction in the body. Inflammatory cytokines and immune cells can alter lipid metabolism and disrupt lipid homeostasis. This disruption can result in changes in lipid profiles, including increased levels of triglycerides and LDL cholesterol and decreased levels of HDL cholesterol [17]. In addition, organ dysfunction or injury caused by COVID-19 can impair lipid metabolism and affect lipid levels in the bloodstream. Less commonly, other features have also been associated with COVID-19 infection, including bile acids [18,19]

Due to time pressure, the novelty of the virus and restricted research capacity, the large majority of omics-based diagnostic tests and pathway analyses did not include independent validation cohorts or follow-up validation [20]. This is important because within-individual variability, between-individual variability and methodological variability in single cohorts can lead to a false discovery of diagnostic biomarkers or misidentification of affected pathways [21]. This is in addition to statistical problems posed by multiple hypothesis testing or the overfitting of models [22].

Furthermore, individual metabolomics studies may not necessarily capture all relevant features or biomarkers in their analysis due to quirks or biases in methods, recruitment cohorts or analysis methodologies. This can be seen by the wide variation in COVID-19 biomarkers identified in the literature. As a simple form of meta-analysis, a data-driven approach to identify the most commonly cited features in the COVID-19 metabolomics literature offers the potential to generate a crowdsourced panel of biomarkers. Crowdsourcing as a method of identifying research parameters has been utilized in a variety of fields [23,24,25], but to our knowledge, no data-driven/crowdsourced panel of biomarkers has yet been investigated for COVID-19.

As the pandemic phase of COVID-19 has receded and the availability of RAT and RT-PCR tests has increased, the immediate requirement for validation of omics methods has also decreased; the validation of methods that are currently in use would naturally be of higher priority [26]. Nonetheless, testing the robustness of the proposed biomarker panels or pathway analyses still has value. This is both to identify best practices for dealing with future pathogens as well as to aid understanding of the biological pathways involved in COVID-19 infection and—for example—how these might influence long COVID, which is an important and ongoing health issue [27,28]. In this work, we used results from a targeted metabolomics study to investigate the robustness of five panels of biomarkers that have been proposed in the literature to be characteristic of COVID-19. We also investigated one panel constructed from the metabolites associated with COVID-19 that were most frequently mentioned in the literature—effectively a crowdsourced biomarker panel. A summary of the workflow is shown in Figure 1.

It should, however, be recognized that this work Is not intended to provide a formal methodological validation of these panels. Nor is it intended to provide analytical comparisons of different omics LC-MS platforms. Method validation for diagnostic tests would best be performed with certified standards with appropriate calibration curves replicating the specific LC-MS modalities employed. Furthermore, testing against a single independent cohort may also introduce bias, especially—as here—when the independent cohort is also small and may not be representative of the wider population, which is a particular issue with a multifactorial disease such as COVID-19. Nonetheless, in this work, we aimed to investigate whether biomarkers associated with COVID-19 could be robustly replicated in a second cohort independent of that used for the original discovery study. We also aimed to identify lessons for study design in the event of future pathogens of concern.

## 2. Results

### 2.1. Biomarker and Panel Selection

The initial literature search for metabolomics panels proposed to be diagnostic or characteristic of COVID-19 infection yielded 31 studies. One of these was selected as an independent dataset, against which the remaining panels could be tested. The independent dataset selected was based on the Biocrates MxP Quant 500 system, using the work of Lewis et al. [19], and it included 472 metabolites across 71 first-wave patient serum samples. This study was chosen because it included a large number of metabolites which were quantitatively measured, using a commercially available platform available to other laboratories; additionally, the full dataset is publicly available. From the remaining 30 studies, 24 were excluded as including one or more biomarkers that could not be replicated in the independent dataset. Of the six studies assessed, one incorporated later dates of sampling when glucocorticoid treatment was widely prevalent, and it was excluded from the analysis due to differing treatment regimes. This left five identified studies with proposed biomarker panels that could also be measured against the independent quantitative dataset used in this work. Including the crowdsourced panel resulted in a total of six panels to be assessed for diagnostic robustness (Table 1).

Reflecting the realities of conducting biomarker research in a pandemic, all five studies were based on retrospective observational cohorts, and—whilst publication ranged from 2020 to 2022—all five studies analyzed first-wave COVID-19 cases.

### 2.2. Testing Performance Metrics for Proposed Biomarker Panels in the Independent Dataset

The summary Area Under the Receiver Operating Characteristic Curve (AUROC) data for each of the panels in Table 1 are shown in Figure 2, and other performance metrics for each of the panels are reported in Table 2. The best performance measured was delivered by Panel 5 (Caterino et al. [33]), with an AUROC of 0.77 and an F1 score of 0.76. The next best performer was Panel 4 (Khodadoust et al. [32]) with AUROC of 0.70 and an F1 score of 0.70. The worst performance measured by AUROC was Panel 1, which was the crowdsourced panel. A ‘control’ measure of AUROC was also generated using 5000 random permutations of five biomarkers, taken from the independent dataset, and then processed in the same way as the other panels. This represents the median value that could be achieved by ‘chance’ through selecting biomarker candidates at random. A histogram and density curve of the AUROC outcomes from the randomly generated biomarker panels is shown in the Appendix A; the median ‘control’ AUROC was 0.58.

Whilst only a small number of studies were included in this work, the correlation between AUROC and the number of features in the panels was tested; there was a positive correlation, but it was not statistically significant (*p*-value of 0.12). As well as AUROC, confusion matrices (true positive, false positive, false negative and true negative) were also constructed for each panel, and these are shown in the Appendix A.

### 2.3. Variations in Performance by Fold

As set out in the Methods section, 5-fold cross-validation was used in this work, such that each 20% fold of the dataset was held out, with the remaining 80% used for training and the 20% fold used for testing. K-fold cross-validation can reduce bias, because all samples are used for both testing and training. Variance can increase between the folds, especially for small datasets, but this can be compensated for by taking the average performance across the folds. For completeness, the performance of each individual fold (i.e., of each of the five 80:20 train:test splits) measured by AUROC is summarized in Figure 3. The crowdsourced panel showed the greatest variation in performance, and Tomo et al. [29] showed the least amount of variation across the folds followed by Caterino et al. [33].

## 3. Discussion

Five biomarker panels were reviewed in this work for robustness and compared with the performance of a crowdsourced panel. The datasets investigated included both sera and plasma as sampling matrices, but overall inter-individual differences have been found to be more marked in metabolite measurement than sample type [34] with concentrations of most metabolites (excepting lactate and citrate) found to be similar between plasma and sera [35]. The biomarkers proposed in the literature panels analyzed in this work were derived from a COVID-19-naive population during the first wave. Subsequent waves were likely to have been confounded by treatment regimes [36], changes in variants [37], and of course by prior exposures/vaccinations [38], all of which can lead to variations in symptoms and/or metabolomic profiles [19]. Whilst these are important confounders, they are beyond the scope of this work to investigate, and so here, we focus on first-wave diagnostic indicators only.

Looking first at the performance of the simpler panels. Tomo et al. proposed that DHEAS and cortisol levels in sera (and the ratio thereof) were associated with COVID-19 [29]. In their work, cortisol was found to be significantly increased (*p* value: <0.001) in COVID-19 cases when compared with controls, and DHEAS decreased (*p* value: <0.001). When tested as a diagnostic panel in independent sera samples, the AUROC of a DHEAS/cortisol-based test was found to be 0.64. These values suggest the association is at best weak, and applying a Student *t*-test to the independent datasets analyzed here found a *p*-value of 0.048 for DHEAS and 0.029 for cortisol; in other words, the order of statistical significance was not robustly replicated in the independent dataset. This may reflect severity, as Tomo et al. also reported that the indicators became more pronounced in worse-affected patients. Nonetheless, as a generalized biomarker panel, DHEAS and cortisol appear only moderately characteristic of COVID-19 infection. Fraser et al. proposed the arginine/kynurenine ratio in plasma as highly diagnostic [30] with a sensitivity of 1.00, specificity of 1.00 and a *p*-value of 0.0002 in their cohort. In the independent dataset, however, this was not reproduced, with an AUROC value of 0.64 and a *p*-value of 0.10 for arginine/kynurenine in the independent dataset. Whilst not a diagnostic proposal, Almulla et al. reviewed 14 articles that compared tryptophan and catabolites in COVID-19 [31], identifying an increase in the kynurenine/tryptophan ratio (*p* value: <0.00001) as a diagnostic of COVID-19. As Almulla et al. used search criteria specifying kynurenine and tryptophan, some bias may have been introduced insofar as studies that focused on other metabolites would not have been included in the meta-analysis. When tested as a diagnostic panel in independent sera samples, the kynurenine/tryptophan ratio was found again to increase, but the AUROC was 0.63 and a Student *t*-test provided a *p*-value of 0.02 in the independent dataset. The ratio still showed statistical significance but to a lesser extent than found in previous studies reporting tryptophan as a key metabolite (and with a less significant *p*-value than the 0.0001 reported by Almulla et al.).

When considering smaller panels overall, the *p*-values reported initially in their respective discovery studies were 0.0002, <0.0001 and <0.00001, but when a Student *t*-test was applied to the same measures in the independent cohort, *p*-values were 0.05, 0.10 and 0.02, respectively, showing a greater-than-order of magnitude difference. Consequently, we find no evidence that simple panels can reproducibly characterize COVID-19 infection.

The second category of biomarker panels included a wider range of indicators. Khodadoust et al. focused only on lipids, rather than amino acids, finding that plasma levels of Cer (d18:0/24:1), Cer (d18:1/24:1), and Cer (d18:1/22:0) were markedly increased in COVID-19-positive patients as compared with healthy controls (and indeed further increased in cases of respiratory distress) [32]. This delivered better performance than the simple panels with an average AUROC of 0.70. Increased ceramide concentrations were also identified in Caterino et al., albeit for different species—Cer (d18:0/20:0), Cer (d18:1/23:0), Cer (d18:1/18:0), Cer (d18:1/26:1) and only for severe cases of infection. Whilst ceramide pathways appear to be dysregulated, the specific lipids affected appear to vary. Ceramides have been associated with immune response and infection and specifically with COVID-19 [39], including via ceramide-enriched membranes’ involvement in cellular infection with SARS-CoV-2 [40]. A dedicated assay targeting ceramides might provide better identification and measurement of this lipid class, and whether it is diagnostic or prognostic, but this is beyond the scope of this work. Whilst Caterino et al. also identified a dysregulation of ceramides in their work, they found a biomarker panel comprising lactic acid, glutamate, aspartate, phenylalanine, β-alanine, ornithine, arachidonic acid, choline, and hypoxanthine offered the best diagnostic accuracy. This set of features produced the best performance amongst those measured here with an AUROC of 0.77. This set also produced a lower-than-average amount of variation in AUROC between folds, which may suggest that the relationship between the markers and COVID-19 status is more consistent than other tests as well as more accurate. The crowdsourced panel, mainly comprising amino acids, performed the least robustly in this independent dataset with an average AUROC of 0.57 and also the widest amount of variation between the folds. Whilst in many fields, the ‘wisdom of crowds’ can outperform single assessments [41,42,43], no such outcome was found in this work. This may reflect the relative ease of metabolite identification. Amino acids require less MS/MS fragmentation for reliable identification than, for example, more complex lipids, leading to their overrepresentation in the literature. It may also reflect a lack of independence in the literature, with later research being influenced by earlier findings [44]. A related form of bias toward prior positive findings has also been seen in other fields with smaller studies with better diagnostic performance being cited more often than later, larger-scale meta-analyses with worse diagnostic performance [45].

Looking beyond possible literature biases, the wide variety of biomarkers identified is strongly suggestive of confounding variables. One such variable is severity with extensive overlap between biomarkers identified as prognostic and diagnostic [9]. Danlos et al. identified reductions in plasma tryptophan levels and increases in levels of the immunosuppressive metabolite kynurenine in critical care patients as compared to mild cases [46]. Similarly, Sindelar et al. found lipids to be prognostic, specifically LPCs and PCs [47]. Chen et al. also highlighted lipids as prognostic, particularly low- and high-density lipoproteins and triglycerides, [48] with Ballout et al. reporting the same prognostic findings [49]. We hypothesize that diagnostic cohorts that focused on more severe positive cases might confuse prognostic biomarkers with diagnostic biomarkers; the latter should show dysregulation even in mild or asymptomatic cases. Other confounding variables can include changes in diet in a hospital environment or length of inactivity as well as between-individual and methodological variation effects [21].

A number of limitations should be highlighted. This work only analyzes a limited subset of biomarkers proposed in the literature. Whilst testing the full range of proposed COVID-19 biomarkers lies beyond the scope of this work, it would be desirable for other panels to be investigated for reproducibility. For example, Delafiori et al. identified a panel comprising a variety of amino acids and lipids with impressive F1 score (0.89), sensitivity (0.83) and specificity (0.96) values [50], but the lipids were not included in the independent dataset used here. Therefore, it is a core limitation of this research that it has not been able to evaluate the full range of proposed diagnostic biomarker panels for COVID-19. Furthermore, this study represents a secondary analysis of existing studies and so inherits the full range of weaknesses of said studies. Sample sizes were small, and due to the observational post hoc nature of recruitment in a pandemic, many variables were not controlled for. For example, the independent cohort used here was recruited in a hospital setting, which may not be representative of other settings. During severe COVID-19 illness, patients may experience prolonged bed rest, which can increase the levels of triglycerides and decrease the levels of HDL cholesterol [51], or diet-related changes in metabolism [16]. Where COVID-19 infections were acquired in hospital settings, it is possible that biases in lipidomic or metabolomic assessments could have occurred. This is particularly the case for the independent cohort used in this work, as any bias or occurrence of confounders such as those described above may lead to the AUROCs for specific biomarker panels being over or under-stated versus that which might be measured in a wider population. This represents a fundamental limitation in this study, and it is to be hoped that future meta-analyses will use larger and/or multiple validation cohorts. It should be remembered, however, that it is desirable for tests for the diagnosis or determination of treatment regime to be resilient to such confounders if they are to be effective in clinical settings.

In this work, all panels (with the exception of the crowdsourced panel) generated an AUROC greater than the ‘control’ value of 0.58 for the independent dataset (the median value achieved by selecting biomarkers at random). Nonetheless, the smaller panels in particular performed worse than originally reported, illustrating the challenge in developing a simple biomarker panel for a multifactorial disease such as COVID-19. This also emphasizes the importance for future pandemics of greater co-operation and the use of independent datasets to improve confidence. Whilst initiatives such as the COVID-19 MS Coalition began the pandemic with good intentions, in practice, two-center studies with the capacity in their testing cohorts to conduct independent—and ideally blind—validation tests were rare. Whilst idealized study design is more difficult in a crisis situation, given that the stakes are also higher, it is to be hoped that collaborative multi-center metabolomics studies will be more widely adopted in any future pandemic.

## 4. Materials and Methods

### 4.1. Dataset and Biomarker Identification

Initially, a review of COVID-19 metabolomics literature was carried out to identify studies proposing panels of metabolites as either diagnostic of COVID-19 or associated with specific dysregulations or pathway alterations. PubMed was searched for articles between January 2020 and June 2023, reviewing all articles covering COVID-19 and metabolomics analyses of participant plasma or sera. Both mass spectrometry (MS) and nuclear magnetic resonance (NMR) studies were considered.

All studies were assessed as to whether they proposed a set of specific metabolites as distinctive of COVID-19, whether explicitly as a diagnostic model or as a signature of the disease. Those studies identifying classes of metabolites, for example identifying triglycerides but not individual features, were excluded. For those studies that met the search criteria, each panel of biomarkers was then assessed to establish whether all featured metabolites were also measured in the independent dataset. Panels where all metabolites were also available in the independent dataset were taken forward for assessment. Panels that were incompletely measured in the independent dataset were not taken forward.

One additional set of biomarkers was considered in this work—a data-driven crowdsourced biomarker panel. This was constructed by scraping PubMed for COVID-19 metabolomics articles, with the search string “COVID-19[Title]) AND ((Metabolism[Title]) OR Metabolites[Title] OR Metabolomics[Title] OR Lipidomics[Title])”, using the R packages easyPubMed [52] and wordcloud2 [53]. A corpus was created using all keywords from articles identified in the search, non-specific words were removed, and metabolite names were lemmatized. A word cloud of the results is included in Figure 1.

### 4.2. Pre-Processing

Once a set of panels had been identified, for each panel, the relevant metabolites were extracted from the independent dataset. These metabolites were log-transformed and unit-scaled. Data visualizations were conducted to gain insights into the distribution of metabolites and identify any potential patterns or relationships. Statistical tests, such as correlation analysis, were performed to assess the significance of associations between the COVID-19 serum metabolites and to identify potential issues with multicollinearity.

### 4.3. Machine Learning Methodology: Supervised Binary Classification

The independent dataset used for this study contained 71 patient samples. For each panel, an 80/20 split for training/testing was used, with 5-fold cross-validation, to facilitate model training and evaluation. Cross-validation was used to avoid bias where the training and test sets are not representative of the overall population, which is an issue for smaller datasets. A logistic regression binary classifier was chosen, having advantages in terms of model interpretability, for example easily understandable feature importance. The disadvantages of logistic regression (poor at dealing with multicollinearity, dependent on linear relationships) were considered to be reduced by the small number of metabolites in each panel.

The training set was used to train multiple logistic regression models, each corresponding to one panel of metabolites (Panels 1–6). Hyperparameter tuning was performed to optimize the logistic regression models’ performance. Grid search cross-validation (CV) was employed to systematically search the hyperparameter space and identify the best combination of hyperparameters. Parameters such as regularization strength (C) and penalty type (l1 or l2) were tuned to find the optimal configuration. Feature importance of the best-performing logistic regression model for each panel was analyzed. This analysis involved evaluating the magnitude and direction of the coefficients assigned to each metabolite. Metabolites with higher absolute coefficient values were considered more important in predicting the binary outcome. Trained logistic regression models were validated using the 20% test set for each fold. Performance for the models comprising Panels 1–6 was evaluated by F1 score, sensitivity, specificity and Youden’s Index (also known as Youden’s J-statistic).

In addition, a control (random) estimate of AUROC was generated by constructing 5000 random panels of five biomarkers taken from the independent dataset. Each was processed by loop using the same logistic regression methodology for the main panels, assessed by AUROC, and the median value of the 5000 panels was taken as a representation of a randomized biomarker panel.

All analyses described herein utilized the Jupyter notebook anaconda version of the Python 3.8.10 programming language to perform the analyses together with the scikit-learn Python package [54]. GridSearch-derived parameters for the trained Logistic Regression classifier are set out in the Appendix A.

## Figures and Tables

**Figure 1 ijms-24-14371-f001:**
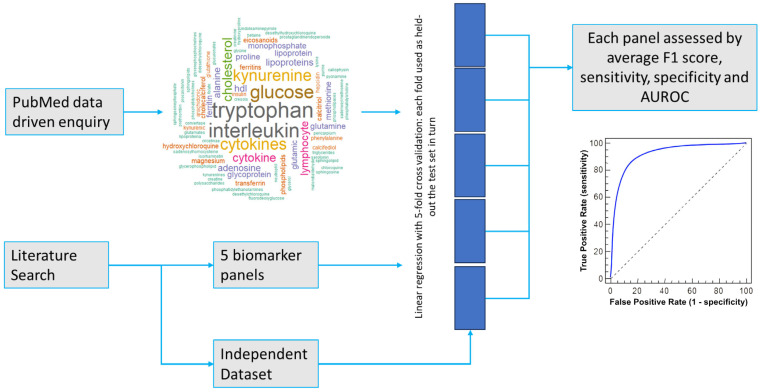
Summary of workflow for testing the robustness of 6 COVID-19 diagnostic biomarker panels in an independent quantitative metabolomic and lipidomic LC-MS dataset.

**Figure 2 ijms-24-14371-f002:**
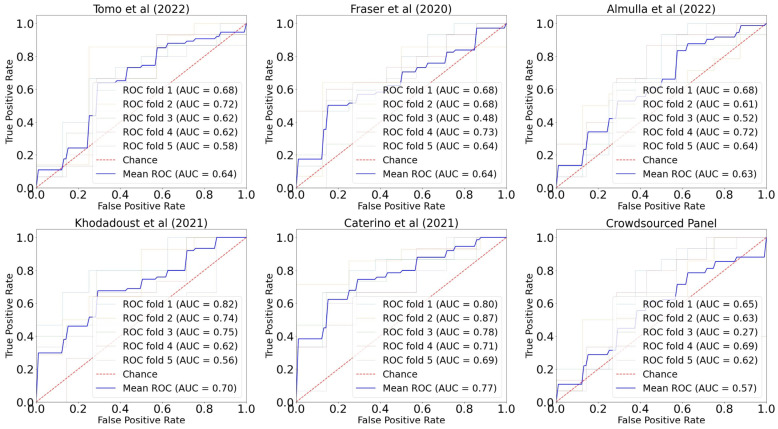
AUROC for each panel [29,30,31,32,33]. Dark blue line shows AUROC averaged across 5 folds. Red dashed line shows the outcome achieved by chance.

**Figure 3 ijms-24-14371-f003:**
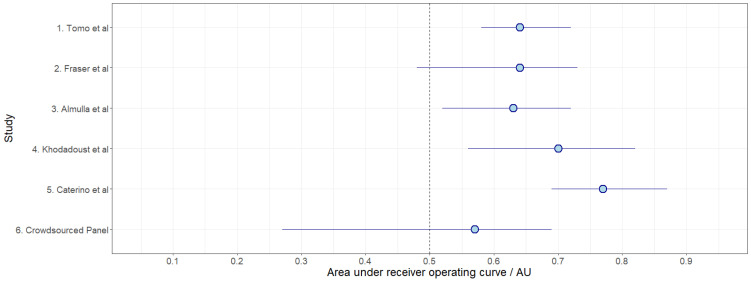
Mean average AUROC for each panel averaged across 5 folds [29,30,31,32,33]. Whiskers show the fold results with the lowest and highest AUROC values.

**Table 1 ijms-24-14371-t001:** Studies identifying metabolomic panels differentiating COVID-19-positive cases from negative either for diagnostic potential or for pathway/biological meaning.

Study	Participants (Positive/Negative)	Sampling Matrix	Identified Biomarkers of COVID-19-Positive Status
1. Tomo et al. (2022) [29]	76/79	Sera	DHEAS, cortisol, DHEAS/cortisol ratio
2. Fraser et al. (2020) [30]	10/10	Plasma	Arginine, kynurenine, arginine/kynurenine ratio
3. Almulla et al. (2022) [31]	329/475 *	n/a	Kynurenine, tryptophan, kynurenine/tryptophan ratio
4. Khodadoust et al. (2021) [32]	60/36	Plasma	Cer (d18:0/24:1), Cer (d18:1/24:1), Cer (d18:1/20:0), Cer (d18:1/22:0)
5. Caterino et al. (2021) [33]	52/9	Sera	Lactic acid, glutamate, aspartate, phenylalanine, β-alanine, ornithine, arachidonic acid, choline, and hypoxanthine
6. Crowdsourced Panel	n/a	n/a	Tryptophan, kynurenine, alanine, glutamine, proline

* Meta-analysis, total number of participants reviewed across multiple studies.

**Table 2 ijms-24-14371-t002:** Performance metrics for serum COVID-19 dataset biomarker metabolites Panels 1–6. Average refers to the mean average performance across the 5 folds.

Panel	Avg. F1 Score	Avg. Sensitivity	Avg. Specificity	Avg. Youden Index	Avg. AUROC
1. Tomo et al. (2022) [29]	0.69	0.62	0.68	0.30	0.64
2. Fraser et al. (2020) [30]	0.70	0.64	0.62	0.25	0.64
3. Almulla et al. (2022) [31]	0.61	0.53	0.59	0.12	0.63
4. Khodadoust et al. (2021) [32]	0.70	0.63	0.67	0.30	0.70
5. Caterino et al. (2021) [33]	0.76	0.72	0.64	0.36	0.77
6. Crowdsourced Panel	0.64	0.57	0.61	0.18	0.57
Control: median result from 5000 random 5-biomarker panels (independent dataset)					0.58

AUROC = Area Under the Receiver Operating Curve; Youden Index = Sensitivity + Specificity − 1.

## Data Availability

The data for the quantitative independent dataset are available at https://zenodo.org/record/7585516, accessed on 6 July 2023.

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
