# Peer review of "Meta-Analysis of COVID-19 Metabolomics Identifies Variations in Robustness of Biomarkers"

_ijms, 2023, doi:10.3390/ijms241814371_

Round 1

Reviewer 1 Report

The authors have presented a meta-analysis on metabolomics in COVID-19 context.

The methodologya nd results are very interesting ad well described. The paper has significant new insights regarding biomarkers in COVID-19. The only concern that I have is the quality of figures, in some figures the fonts are so small that one can not read them.

Author Response

Meta-analysis of COVID-19 metabolomics identifies variations in robustness of biomarkers

We thank the reviewers for their helpful and constructive comments, and have included our responses below. Where line numbers to the revised text are given, these line numbers refer to the text with ‘track changes’ on.

REVIEWER ONE

The authors have presented a meta-analysis on metabolomics in COVID-19 context.

The methodology and results are very interesting and well described. The paper has significant new insights regarding biomarkers in COVID-19. The only concern that I have is the quality of figures, in some figures the fonts are so small that one can not read them.

We thank the reviewer for their positive comments. We have increased the font size as much as possible in the Figures to improve readability.

REVIEWER TWO

The study by Onoja et al is a meta-analysis investigating the robustness of biomarkers of COVID-19 derived from metabolomics studies. Authors validate 5 previously published biomarker panels and a novel panel based on pubmed driven enquire (crowd-sourced panel) finding significant differences between them. Although I believe that such meta studies are important and still relevant, I believe the report by Onoja et al has significant flaws precluding clear-cut conclusions. Out of 31 studies, they could only evaluate 5. The evaluation was furthermore based on a single “test” study and the reasons for choosing this single study are not given in detail. Furthermore, relying on a single test study could introduce significant bias and thus distort conclusions. Therefore, although the approach presented is valid in principle, , a lot more work needs to be done to ensure confounding factors are excluded and I believe that the data is not sufficient to draw the conclusions as presented.

We agree with the reviewer that the study has clear limitations, and have attempted to address this and other issues as per the specific replies to the detailed comments 1 to 5 below.

Below I provide further clarifications.

  1. Researchers start with 31 studies for their meta-analysis and one study as a test dataset. Often times it is very difficult to compare metabolomics studies across labs in part due to the nature of LC-MS measurements. Researchers need to provide further detail on their choice for a test study. Was the test study one with largest number of metabolites (and/or participants) measured? Why was this particular study appropriate as a test study? Further, researchers should select at least a second independent test study and look for overlaps. In addition, researchers would need to explore similarities between studies, such as particular platforms (Metabolon vs Biocrates vs in-house, TOF vs Orbitrap vs QQQ etc etc) used and how that might affect conclusions. A parallel approach could also be to allow for missing metabolites but rather pair re-evaluated studies with a more suitable test study. All in all, I believe very little can be reliably learned from the investigation as currently designed. An inability to reproduce previous results (biomarker validation) could stem from multiple factors that researchers cannot account for.

The test study was chosen because it provided the full dataset for public access (many studies do not) and had the widest range of metabolites, therefore allowing the widest range of panels to be tested against it. We have amended the text to make this clear. We agree that it would be desirable to explore studies using different platforms, such as Metabolon vs Biocrates vs in-house, TOF vs Orbitrap vs QQQ. But in our view, this would be a methods validation paper, rather than a meta-analysis of panel performance. Datasets to allow comparison of all these different methods are not in the public domain, and such a work is well beyond the scope of this manuscript. We have amended the text to reflect these concerns [lines 91-92], including expanding the limitations paragraph [lines 273 to 288].

Finally, whilst we acknowledge that there may be issues in the independent dataset that mean that there could be multiple factors explaining the reduction in diagnostic power of the literature panels, we think this is still interesting. To be reliable (e.g. in a clinical setting), panels for diagnosis or for treatment determination should be robust to confounders such as different cohorts. If panels are shown to be less reliable in an independent dataset, this is in our view worthy of reporting. We have edited our manuscript to ensure the emphasis lies upon this observation [lines 28-29 and 286-288].

  1. Many of the studies that researchers investigate, including the test study, are based on very few participants. This is a major limitation. More recent studies could be included that report on vastly higher number of participants. This is especially true for the test study. Furthermore, there could be a lot of confounding factors, such as race, age, or income levels – all factors that can influence disease progression as well as metabolic status. Therefore, a test study with large number of participants, for which these factors could be controlled, would be preferred.

As above, we agree that a test study with more participants would be preferred. As a meta-analysis, we have used those studies with their data in the public domain. Larger studies have been published but often do not make their full data available, for example only reporting data for key markers, preventing their use for validation of other biomarker panels. The expanded limitations paragraph [lines 273 to 288] also covers this point.

Minor comments:

  1. Researchers could test in addition to their “crowd sourced” metabolite set, also a set of random metabolites as control.

We thank the reviewer for this excellent suggestion, which we have incorporated [lines 132-137 and 355-359]. Rather than a single set of random metabolites, which could randomly be ‘good’ or ‘bad’, we have generated a series of random metabolite panels and selected the median as a “control” AuC metric. The text is amended accordingly and a histogram of the AuC’s achieved in the random panels is also included in Supplementary Material, Figure S2. The median AuC was 0.58, which naturally has been skewed away from 0.50 by the presence of diagnostic markers.

  1. For table 2 – a column with the original reported score per study could be informative

Unfortunately, the papers reviewed did not report original scores consistently, some using p-values, some using fold changes, some using accuracy. We have included additional commentary in the text for each study, for example lines 216-221.

  1. Discussion could be shortened overall and the paragraph on limitations could be expanded.

We have expanded the Limitations paragraph and edited Discussion to remove any extraneous text, We hope that this addresses the reviewers concerns.

Reviewer 2 Report

The study by Onoja et al is a meta-analysis investigating the robustness of biomarkers of COVID-19 derived from metabolomics studies. Authors validate 5 previously published biomarker panels and a novel panel based on pubmed driven enquire (crowd-sourced panel) finding significant differences between them. Although I believe that such meta studies are important and still relevant, I believe the report by Onoja et al has significant flaws precluding clear-cut conclusions. Out of 31 studies, they could only evaluate 5. The evaluation was furthermore based on a single “test” study and the reasons for choosing this single study are not given in detail. Furthermore, relying on a single test study could introduce significant bias and thus distort conclusions. Therefore, although the approach presented is valid in principle, , a lot more work needs to be done to ensure confounding factors are excluded and I believe that the data is not sufficient to draw the conclusions as presented. 

Below I provide further clarifications.

1. Researchers start with 31 studies for their meta-analysis and one study as a test dataset. Often times it is very difficult to compare metabolomics studies across labs in part due to the nature of LC-MS measurements. Researchers need to provide further detail on their choice for a test study. Was the test study one with largest number of metabolites (and/or participants) measured? Why was this particular study appropriate as a test study? Further, researchers should select at least a second independent test study and look for overlaps. In addition, researchers would need to explore similarities between studies, such as particular platforms (Metabolon vs Biocrates vs in-house, TOF vs Orbitrap vs QQQ etc etc) used and how that might affect conclusions. A parallel approach could also be to allow for missing metabolites but rather pair re-evaluated studies with a more suitable test study. All in all, I believe very little can be reliably learned from the investigation as currently designed. An inability to reproduce previous results (biomarker validation) could stem from multiple factors that researchers cannot account for.

2.Many of the studies that researchers investigate, including the test study, are based on very few participants. This is a major limitation. More recent studies could be included that report on vastly higher number of participants. This is especially true for the test study. Furthermore, there could be a lot of confounding factors, such as race, age, or income levels – all factors that can influence disease progression as well as metabolic status. Therefore, a test study with large number of participants, for which these factors could be controlled, would be preferred.

Minor comments:

3. researchers could test in addition to their “crowd sourced” metabolite set, also a set of random metabolites as control.

4. for table 2 – a column with the original reported score per study could be informative

5. Discussion could be shortened overall and the paragraph on limitations could be expanded.

no specific comments

Author Response

(The authors gave the same response as above.)

Round 2

Reviewer 2 Report

I have reviewed author's responses and although I agree that the study could still merit publication and be informative I have remaining concerns and would not recommend publication in the current form. However, my concerns  could still be addressed simply. Specifically, I do not find the changes suggested on my comment 1 to be sufficient. The amended text (line 91-92 and line 273 to 288) is not addressing the comment adequately. Actually, I do not see lines 273-288 being changed at all and if the line reference is wrong I don't find relevant text in discussion. I would advise authors clearly state the limitations of their study in main text as well as in discussion. As well as clearly comment on why the use of a single test study could create bias.

Dear Editor, although I agree that the study as presented could still merit publication and be informative I do not believe my comments were sufficiently addressed. Please, see my response to authors. I have suggested additional changes that are on the level of simple addition of text to the manuscript.

Author Response

Dear reviewer,

Thank you for the comments. We agree that the text will benefit from emphasising the limitations of the current study and have revised the manuscript, and have pasted in the updated paragraphs below for your convenience.

INTRODUCTION

It should, however, be recognized that this work is not intended to provide formal methodological validation of these panels. Method validation for diagnostic tests would best be performed with certified standards with appropriate calibration curves replicating the specific LC-MS modalities employed. Furthermore, testing against a single independent cohort may also introduce bias, especially – as here – when the independent cohort is also small and may not be representative of the wider population, a particular issue with a multifactorial disease such as COVID-19. Nonetheless, in this work we aimed to investigate whether biomarkers associated with COVID-19 could be robustly replicated in a second cohort independent of that used for the original discovery study. We also aimed to identify lessons for study design in the event of future pathogens of concern.

LIMITATIONS

A number of limitations should be highlighted. This work only analyses a limited subset of literature proposed biomarkers. Whilst testing the full range of proposed COVID-19 biomarkers lies beyond the scope of this work, it would be desirable for other panels to be investigated for reproducibility. For example, Delafiori et al identified a panel comprising a variety of amino acids and lipids with impressive F1 score (0.89), sensitivity (0.83) and specificity (0.96), [50] but the lipids were not included in the independent dataset used here. Therefore, it is a core limitation of this research that it has not been able to evaluate the full range of proposed diagnostic biomarker panels for COVID-19. Furthermore, this study represents a secondary analysis of existing studies, and therefore this work inherits the full range of weaknesses of said studies. Sample sizes were small, and due to the observational post-hoc nature of recruitment in a pandemic, many variables were not controlled for. For example, the independent cohort used here was recruited in a hospital setting, which may not be representative of other settings. During severe COVID-19 illness, patients may experience prolonged bed rest, which can increase the levels of triglycerides and decrease the levels of HDL cholesterol, [51] or diet-related changes in metabolism. [16] Where COVID-19 infections were acquired in hospital settings, it is possible that biases in lipidomic or metabolomic assessments could have occurred. This is particularly the case for the independent cohort used in this work, as any bias or occurrence of confounders such as those described above may lead to the AUROCs for specific biomarker panels being over or under-stated versus that which might be measured in a wider population. This represents a fundamental limitation in this work, and it is to be hoped that future meta-analyses will use larger and / or multiple validation cohorts. It should be remembered, however, that it is desirable for diagnostic tests to be resilient to such confounders if they are to be effective in clinical or home settings.